# Gastroparesis and Dumping Syndrome: Current Concepts and Management

**DOI:** 10.3390/jcm8081127

**Published:** 2019-07-29

**Authors:** Stephan R. Vavricka, Thomas Greuter

**Affiliations:** 1Center of Gastroenterology and Hepatology, CH-8048 Zurich, Switzerland; 2Department of Gastroenterology and Hepatology, University Hospital Zurich, CH-8091 Zurich, Switzerland

**Keywords:** gastroparesis, dumping syndrome, pathophysiology, clinical presentation, treatment

## Abstract

Gastroparesis and dumping syndrome both evolve from a disturbed gastric emptying mechanism. Although gastroparesis results from delayed gastric emptying and dumping syndrome from accelerated emptying of the stomach, the two entities share several similarities among which are an underestimated prevalence, considerable impairment of quality of life, the need for a multidisciplinary team setting, and a step-up treatment approach. In the following review, we will present an overview of the most important clinical aspects of gastroparesis and dumping syndrome including epidemiology, pathophysiology, presentation, and diagnostics. Finally, we highlight promising therapeutic options that might be available in the future.

## 1. Introduction

Gastroparesis and dumping syndrome both evolve from a disturbed gastric emptying mechanism. While gastroparesis results from significantly delayed gastric emptying, dumping syndrome is a consequence of increased flux of food into the small bowel [1,2]. The two entities share several important similarities: (i) gastroparesis and dumping syndrome are frequent, but also frequently overlooked; (ii) they affect patient’s quality of life considerably due to possibly debilitating symptoms; (iii) patients should be taken care of within a multidisciplinary team setting; and (iv) treatment should follow a step-up approach from dietary modifications and patient education to pharmacological interventions and, finally, surgical procedures and/or enteral feeding. Most importantly, the two diagnoses have to be considered by one of the treating specialists, regardless of whether this is the endocrinologist, nutritional specialist or gastroenterologist, when symptoms are present. Pre-test probability based on comorbidities (such as diabetes in case of gastroparesis or surgical history for dumping syndrome) together with the presence of typical symptoms should lead to a high degree of clinical suspicion. However, for both disorders, diagnostic evaluations should follow in order to confirm the diagnosis before initiation of treatment. Firstly, because treatment options might be invasive and require proper diagnostic evaluations beforehand. Secondly, several differential diagnoses might show a similar presentation. Such diagnoses are peptic ulcer disease, gastric cancer, celiac disease, abdominal angina for gastroparesis, anastomotic ulcers, internal herniation and gallbladder disease for early dumping syndrome and insulinoma, surreptitious use of glucose-lowering medication for late dumping [2,3,4,5]. In the following review, we will present an overview of the most important clinical aspects of gastroparesis and dumping syndrome including epidemiology, pathophysiology, presentation, diagnostics and treatment. Finally, we highlight promising therapeutic options that might be available in the future.

## 2. Definitions and Epidemiology

Gastroparesis and dumping syndrome are frequent, but their prevalence and incidence vary depending on definitions and studied populations. Therefore, heterogenous results have been reported in the literature.

### 2.1. Gastroparesis

Gastroparesis is a syndrome characterized by an objectively delayed gastric emptying in the absence of a mechanical gastric outlet obstruction and the presence of cardinal symptoms such as early satiety, postprandial fullness and nausea-vomiting [6]. The prevalence of gastroparesis in the general population is uncertain. A wide range in different at-risk populations has been reported. In addition, gastroparesis is likely significantly under diagnosed. While an epidemiological study from Olmsted county revealed a prevalence of 24.2/100,000 for definite gastroparesis and 50.5/100,000 for definite, probable or possible gastroparesis [7], prevalence might be as high as 1.8% [8]. Patients with type 1 diabetes are at particular risk. Here, 10-year incidence rates of 5.2% have been reported (in contrast to a rate of 1% for type 2 diabetes and 0.2% for non-diabetic patients [9]. Other studies demonstrate even higher rates for diabetics with 58% for type 1 and 30% for type 2 [10,11]. However, most of the performed studies have a considerable selection bias with inclusion of patients from tertiary referral centers only. Still, there might be a large proportion of undetected gastroparesis patients, because either the patient does not seek medical attention or the treating doctors are reluctant to evaluate symptoms and/or further diagnostics. The incidence of postsurgical gastroparesis after gastrectomy is approximately 0.4% to 5.0% [12]. Overall, the incidence of gastroparesis after surgery depends on the surgical procedure and the surgical site. In the early postoperative period after pylorus-preserving pancreatoduodenectomy, postsurgical gastroparesis occurs in up to 20% to 50% of patients [12]. In one study, 67% of patients who underwent pancreatic cancer cryoablation were found to suffer from gastroparesis [13]. There seems to be a gender-specific differences with women accounting for up to 70% of the affected population. Interestingly, elderly patients (>65 years old) are at particular risk [14].

### 2.2. Dumping Syndrome

Dumping syndrome is a frequently encountered postsurgical complication that can be divided into an early and late subtype [2]. Alterations in gastric anatomy after esophageal, gastric and bariatric surgery result in rapid passage of food into the small intestine, which leads to early gastrointestinal and vasomotor symptoms (within 1 h) and late hypoglycemia (1 to 3 h after meal ingestion) [15,16]. Reliable population-based prevalence data for dumping syndrome are still lacking. As of yet, the frequency of postsurgical dumping syndrome is estimated at 25%–50% with 5 to 10% of patients experiencing a severe disabling form [17]. These rates vary depending on type of surgery prodecure [2]. While 20% of patients suffer from symptoms of dumping syndrome after vagotomy and pyloroplasty, these rates rise to 40% after Roux-en-Y bypass and sleeve gastrectomy, and peak at 50% after esophagectomy [18,19,20,21,22]. The incidence and prevalence of dumping syndrome has been increasing due to the current obesity epidemics and the consecutive climb in gastric bypass surgeries [23]. Early dumping represents the most common type, while isolated late dumping is observed in only 25% [2,22]. Due to considerable overlap in clinical presentation it is, however, sometimes difficult to differentiate between the two and co-occurrence is frequently encountered.

## 3. Pathophysiology and Clinical Presentation

The occurrence of cardinal symptoms after ingestion of a meal in a patient with high pre-test probability should rise suspicion for the presence of gastroparesis or dumping syndrome. The symptoms per se are rather non-specific and might occur with many other diseases. However, the existence of risk factors such as diabetes for gastroparesis or bariatric surgery for dumping makes the diagnoses more likely (Table 1 and Table 2).

### 3.1. Gastroparesis

Several aspects contribute to a delayed gastric emptying in gastroparesis patients. Among these are extrinsic denervation of the stomach, impaired inhibitory input to smooth muscles due to loss of nitric oxide in enteric nerves, loss of interstitial cells of Cajal (ICC, “pacemaker cells), smooth muscle atrophy and altered function of immune cells [1]. ICC generate electrical slow waves in the stomach, and disrupted ICC networks and gastric dysrhythmias have been associated with gastroparesis [24]. For details see Figure 1. The most frequent etiologies are diabetes and surgery [25]. In a high proportion of patients, the underlying cause remains unknown (so called idiopathic gastroparesis) [25]. This form is found in particular in younger women and appears to be associated with viral infection (in up to 20%) [26,27,28]. Less frequent etiologic factors are Parkinson’s disease, amyloid, tumors (paraneoplastic gastroparesis), scleroderma, or mesenteric ischemia [29]. Importantly, medication-induced gastroparesis has to be considered in all patients. Typically here are opioids, ciclosporine, anticholinergics and glucagon-like peptide 1 (GLP1)-agonists [30,31,32]. The latter should be particularly suspected in diabetic patients before considering the gastroparesis to be caused by diabetes and poor control of blood glucose. From a clinical perspective, the simplest classification of gastroparesis is into the two categories diabetic vs. non-diabetic. Delayed gastric emptying can be observed in 28% to 65% of unselected patients with diabetes [33,34]. Patients with type 1 diabetes have a higher incidence of gastroparesis as compared with type 2 (5.25 vs. 1%) and an earlier age at onset [7,9,29]. Type 2 diabetics, however, have more serious symptoms. Gastroparesis in diabetic patients usually occurs 10 years after the onset of diabetes and it parallels other forms of diabetic microvascular disease, including neuropathy and retinopathy. Severe symptoms of diabetic gastroparesis cause poor glycemic control and poor nutritional status, and increase the risk of hypoglycaemia [35].

Gastroparesis is often a debilitating disease associated with significant morbidity and mortality [36,37]. The most frequently reported symptoms are: early satiety, postprandial fullness, nausea-vomiting, bloating and upper abdominal pain [38]. The latter is somewhat neglected by physicians, but appear to occur in up to 72%–90% of patients. Pain is often experienced after meals (72%), but can also occur during night time (74%) [29,39]. The presence of non-prandial abdominal pain can, therefore, not be used to rule out the diagnosis. It rather appears to be another piece in the wide presentation of the disease. Weight loss is not a typical feature of gastroparesis patients. Symptoms can vary depending on the specific etiology. Nausea, at least, is similar in all patients with gastroparesis, irrespectively of the underlying cause. Both nausea and vomiting significantly correlate with a reduced quality of life [40]. It should be kept in mind that the presence of symptoms is a sine qua non for the diagnosis of gastroparesis.

### 3.2. Dumping Syndrome

In early dumping syndrome, rapid transition of food into the small intestine results in a fluid shift (due to hyperosmolarity) and release of gastrointestinal hormones such as vasoactive substances (neurostatin, vasoactive intestinal peptide (VIP), incretins (gastric inhibitory polypeptide (GIP), GLP1) and glucose-modulators (insulin, glucagon) [2,15,41,42,43]. This results in gastrointestinal and vasomotor symptoms; the latter is characterized by hypotension (due to the above mentioned fluid shift) and a consecutive sympathetic nerve system response [15,16]. In late dumping, rapid absorption of glucose (due to the presence of undigested carbohydrates in the small intestine) increases incretin release (GLP1), which exaggerates insulin response [2,44,45,46]. This ultimately leads to hypoglycaemia within 1 to 3 h after meal intake (so called incretin-driven hyperinsulinemic response) [15,16]. Although these mechanisms are typically and frequently seen as a postsurgical complication, dumping can also occur in the absence of previous surgery [47]. While in these cases, there is no alteration in gastric anatomy, gastric emptying is accelerated due to a disturbed intrinsic gastric innervation [2]. Diabetes is the main contributor [47,48]. Thus, diabetes can be associated with both delayed (see gastroparesis) and increased gastric emptying. The latter is particularly seen in early diabetes and more so in type 2 than 1. Due to overlapping symptoms, it might be difficult to distinguish gastroparesis from dumping syndrome in diabetic patients. Vomiting makes gastroparesis more probable, while dumping should be considered in particular in the presence of diarrhea [49]. Idiopathic forms of dumping have been described also. Similarly to gastroparesis, viral infections are a possible cause [50]. About 50% of patients with idiopathic dumping syndrome report a history of gastroenteritis symptoms [50].

Symptoms of early and late dumping are quite different. Early dumping includes the following symptoms that occur within 1 h, typically 30 min after a meal ingestion [15]: abdominal pain, bloating, borborygmi, nausea, diarrhea (gastrointestinal symptoms); and fatigue, desire to lie down, flushing, palpitations/perspiration, tachycardia, hypotension and rarely syncope (vasomotor symptoms). For late dumping, typical symptoms can be divided into neuroglycopenia and autonomic reactivity [17]. While the first group comprises fatigue, weakness, confusion, hunger and syncope, the latter includes perspiration, palpitations, tremor and irritability. Dumping syndrome can result in significant weight loss (30%) and considerably affects quality of life [51]. It may lead to hospitalizations due to hypoglycemia, episodes of confusion, seizures and even epilepsy [52].

## 4. Diagnostic Evaluations

Gastroparesis and dumping syndrome have to be distinguished from several organic and functional diseases. Distinction between irritable bowel syndrome, functional dyspepsia and gastroparesis, dumping syndrome might be particularly difficult [29,53]. Many cardinal symptoms are shared between these entities. Diagnostic evaluations are therefore key to clearly establish the diagnosis of gastroparesis and dumping syndrome [54].

### 4.1. Gastroparesis

Gastroparesis is diagnosed by a thorough history and physical examination, which help to identify the presence of characteristic symptoms of gastroparesis and exclude alternative diagnoses. Once the syndrome is suspected, an upper endoscopy is necessary to rule out mechanical gastric obstruction. Although an upper gastrointestinal series may suggest the diagnosis, there is a limited role for imaging in the workup for gastroparesis and delayed emptying should be confirmed by gastric transit testing such as gastric scintigraphy, breath testing, and wireless motility capsule [55,56]. Gastroparesis is diagnosed if typical symptoms are present, a delayed gastric emptying is documented and a correlation of symptoms with food intake is observed [29]. This definition of gastroparesis highlights two key points: (1) objective measures of delayed gastric emptying are needed to establish the diagnosis; and (2) symptoms are a sine-qua-non [29]. A test that is positive for delayed gastric emptying does not make the diagnosis if a patient is asymptomatic. Scintigraphy is the gold-standard to establish decreased gastric emptying and the most widely used technique. However, there are regional differences. Some European centers use breath tests more often because there are simply to handle and have been shown to correlate nicely with scintigraphy results [57]: A test meal (usually white bread, butter, water and an egg omelette with 13C octanoic acid) is ingested and 13C measured using expired breath gas analysis [58]. Magnetic resonance imaging (MRI) assesses gastric physiology by simultaneously measuring gastric emptying and motility without the need of ionizing radiation [59]. There is increasing evidence that MRI is a reliable non-invasive tool in the diagnostic evaluation of gastroparesis [59]. Given the importance of disrupted ICC networks in the pathophysiology of gastroparesis, slow wave measurements are another potential diagnostic test. In fact, a combination of electrogastrogram and magnetogastrogram is able to distinguish gastroparesis patients from controls [60].

### 4.2. Dumping Syndrome

Increased pre-test probability such as a past surgical history and the presence of typical symptoms result in a high clinical suspicion for the presence of dumping syndrome. Two questionnaires have been established for the identification of clinically relevant symptoms of dumping syndrome, the Sigstad’s score and the Arts’ dumping questionnaire [2,51]. A simple visual analogue scale might also be used [61]. Objective measurements are less established compared to gastroparesis. Measuring glucose has a low diagnostic yield, but might be considered when symptoms of late dumping syndrome are present [2]. Importantly, glucose should be measured in the plasma and not in the capillary blood, since low glucose levels are less reliable in the latter. Suggested cut-offs are 2.8 and 3.3 mmol/L [62]. Provocation tests might have higher predictive values. Two options are available, oral glucose tolerance test (OGTT) and the mixed meal tolerance test [2,63]. For OGTT, 50 or 75 g sugar is ingested and blood glucose, haematocrit, pulse and blood pressure are measured every 30 min for 3 h [15]. An increase in the haematocrit by 3% and/or pulse acceleration by 10 min suggest early dumping syndrome, while development of hypoglycemia suggests late dumping syndrome [15]. In the mixed meal tolerance test, carbohydrates, fats and proteins are ingested, and glucose, insulin are checked every 30 min [44,64]. The ordering physician should keep in mind that even healthy individuals can show a drop in their blood sugar levels after meal intake and that therefore the test has a high false positive rate [65]. Gastric emptying studies are potential alternatives, but their specificity and sensitivity is rather low [2].

## 5. Treatment of Gastroparesis

Treatment strategies are quite similar in gastroparesis and dumping syndrome. In a multidisciplinary setting, a step-wise approach should be followed, where dietary modifications and patient education represent the first step, pharmacological interventions the second, and surgical interventions the last option (Figure 2). In the acute setting of gastroparesis, dehydration and electrolyte abnormalities should be corrected by oral or intravenous routes, as appropriate [66]. In severe cases, gastric decompression by insertion of a nasogastric tube might be necessary [29]. In a more chronic setting, the first step in the management of gastroparesis are nutritional counselling and dietary modifications. The patient should be educated to eat small, low-fat, low-fibre meals [29]. Counselling by a nutritional expert is of paramount importance, because gastroparesis can lead to considerable malnutrition due to inadequate oral intake and vomiting [67,68]. In case of underlying diabetes, controlling blood glucose levels should be aimed for [29]. Epidemiological studies have linked poor diabetes control with gastroparesis [69]. As of yet, there are no randomized controlled trials proving that lowering HbA1c indeed improves gastroparesis symptoms. Nonetheless, controlling diabetes makes sense for numerous reasons. When these initial steps do not lead to clinical improvement, pharmacological interventions are needed. The armamentarium is limited, but at least a few drugs are available [70]. Among those are the prokinetics metoclopramide and domperidone that act as dopamine 2 receptor antagonists. Their efficacy has been demonstrated in several randomized trials [71,72]. However, their efficacy appears to be independent of their potential for accelerating gastric emptying, since gastric emptying has been shown to poorly correlate with symptomatic response [73]. Other mechanisms such as affecting gastric hypersensivity or gastric accommodation may be responsible for their effect [74]. Data about their long-term effects are sparse, and several potentially deleterious side-effects limit their use in clinical practice. Metoclopramide imposes a risk for tardive dyskinesia in 1% of patients, while domperidone has been linked to serious cardiac side-effects. Use of metoclopramide should, therefore, be restricted to a maximum of 3 months, at the lowest possible dose [75]. Erythromycin, a macrolide antibiotic, improves gastric emptying through stimulating the motilin receptor and represents an option for short-term treatment, when metoclopramide and domperidone have failed [76]. Nausea, a frequent and bothersome symptom in gastroparesis, can be treated with antiemetics such as prochlorperazine, diphenhydramine, or a 5-HT3 antagonist [66]. Aprepitant can also be considered given its effects on nausea, vomiting and overall symptoms in gastroparesis patients [77]. Iberogast, an over-the-counter herbal preparation is frequently used in functional dyspepsia. Despite having no effect on gastric emptying, iberogast has been shown to positively affect gastroparesis symptoms in a placebo-controlled crossover trial [78]. Several interventional approaches have been developed and hyped in the past with different outcomes. The injection of botulinum toxin during upper endoscopy into the pyloric muscle might be effective, albeit only in the short term [79,80,81,82]. However, the two only randomized controlled trials failed to show efficacy of pyloric botox [83,84]. Whether this was due to small sample size and consecutive lack of power has yet to be determined. Another therapy option is gastric electrical stimulation (GES), where electric current is delivered to gastric smooth muscles via implanted electrodes with a positive impact on gastroparesis symptoms and gastric emptying [85]. GES has been proposed as an alternative for patients with intractable gastroparesis, but a double-blind controlled trial showed only minor improvement with a significant complication rate [86]. A meta-analysis, including 10 studies, suggested that diabetic gastroparesis patients seem to benefit the most, whereas idiopathic gastroparesis patients and postsurgical patients are less responsive and need further research [87]. Another technique, the gastric peroral endoscopic pyloromyotomy (G-POEM) showed some efficacy in gastroparesis patients but further studies are needed [88,89,90,91,92,93]. Esoflip is currently the newest kid on the block. It is a balloon catheter that was developed for dilations in the gastrointestinal tract. Non-controlled studies revealed high rates of technical success with increased distensibility and symptomatic improvement after pyloric dialation [94]. Esoflip might have its role in difficult to treat patients before considering surgical interventions, but more data, particularly randomized controlled trials are needed first. Surgery should be seen as treatment of last resort and should be discussed individually in a multidisciplinary team setting due to the sparse clinical data. At least, a recent case series including 28 patients suggested improvement of symptoms, gastric emptying and a consecutive reduction in the need for prokinetic treatment 3 months after surgery [95]. However, surgery clearly has its role when oral nutritional intake is compromised. In these cases, jejunostomy feeding should be considered. A percutaneous or surgical placement of a gastrostomy-jejunostomy tube enables decompression of the stomach and permits enteral feeding [96].

## 6. Treatment of Dumping Syndrome

Similarly to gastroparesis, dietary modifications and patient education by a nutritional expert are the first steps in the treatment of dumping syndrome [2,97,98,99,100]. Smaller and more frequent meals (around six per day) are recommended [2]. Intake of fluids should be delayed by at least 30 min. Rapidly absorbable carbohydrates and alcoholic beverages should be avoided, while intake of high-fibre, high-protein food is recommended [2]. The rationale behind is that liquids further accelerate gastric emptying. Lying down after a meal for 30 min may further delay emptying of the stomach [15,101]. As a second step, dietary supplements such as guar gum, pectin or glucomannan can be added to enhance food viscosity [102,103,104]. However, most patients do not tolerate these agents due to frequent side-effects such as gas formation and bloating [2]. Pharmacological interventions are the third step in the therapeutic ladder. However, it should be kept in mind that as of yet no treatment has been approved for dumping syndrome. The currently available options are (1) acarbose and (2) for severe cases somatostatin analogues. Acarbose is an alpha-glucosidase inhibitor that decreases intraluminal digestion of carbohydrates in the duodenum. Therefore, it is used to treat postprandial hypoglycemia in late dumping syndrome [105,106,107,108]. However, as seen for the dietary supplements, acarbose is associated with side-effects such as flatulence occurring in a high proportion of patients [109]. Somatostatin analogues can improve dumping syndrome by affecting several disease mechanisms: (i) delaying gastric emptying; (ii) slowering small intestine transit; (iii) decreasing release of gastrointestinal hormones including insulin secretion; and (iv) inhibition of postprandial vasodilation [2,51]. Somatostatin analogues such as octreotide can be applied either subcutaneously three times a day or intramuscularly every 2 to 4 weeks [51,110,111,112]. The latter is probably preferred by the majority of patients. Most important side-effects are steatorrhea, diarrhea, nausea, gallstone formation, pain at injection site and weight gain. Due to these side-effects daily applied doses should start as low as 25 mcg and then titrated up to a maximum of 100 mcg. Treatment should be stopped after 2 weeks (subcutaneous application) or 2 months (intramuscular application) if no improvement is observed [2]. Non-specific symptoms such as nausea and diarrhea may be symptomatically treated with antiemetics (meclizine, promethazine) and antidiarrheals (tincture of opium, loperamide). Anticholinergics like dicyclomine, hyoscyamine, and propantheline slow gastric emptying and are antispasmotic. Diazoxide is a potassium channel activator, which influences hypoglycaemia and is used in late dumping [113]. However, these drugs have been studied in only few patients with dumping syndrome. As for gastroparesis, surgical interventions (revision surgery) should be discussed on a case to case basis, since data for their efficacy are limited [2].

## 7. Conclusions

Gastroparesis and dumping syndrome are frequent disorders, particularly in diabetic and postsurgical patients. Since symptoms are non-specific, these two entities have to be distinguished from several differential diagnoses and require objective measures that document delayed or accelerated gastric emptying. A multidisciplinary team approach including nutritional experts, endocrinologists and gastroenterologists is key to success. Within this setting, optimal treatment strategies can be discussed and individually tailored to each patient. This is particularly important in cases that are difficult to treat, where possible treatment options are limited and data for their efficacy conflicting.

## Figures and Tables

**Figure 1 jcm-08-01127-f001:**
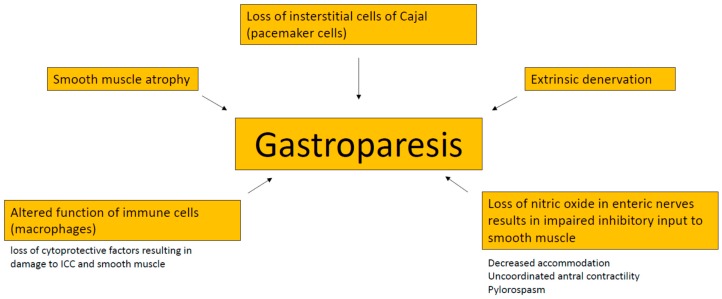
Pathophysiology of gastroparesis. ICC, interstitial cells of Cajal.

**Figure 2 jcm-08-01127-f002:**
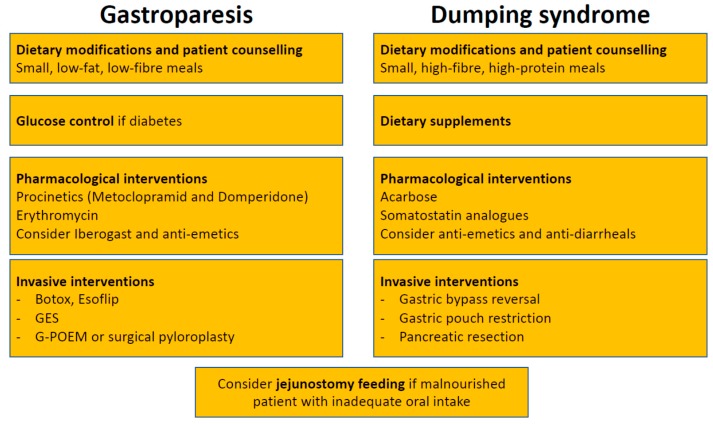
Treatment algorithm for gastroparesis and dumping syndrome. GES, gastric electrical stimulation; G-POEM, gastric peroral endoscopic pyloromyotomy.

**Table 1 jcm-08-01127-t001:** Symptoms and causes of gastroparesis.

Symptoms and Causes of Gastroparesis
Symptoms:
Postprandial fullness, early satiety, bloating, abdominal distension, nausea, and vomiting, abdominal pain, and dysphagia
Causes:
Diabetes Non-diabetic causes Surgery (such as gastrectomy with vagotomy, vagal nerve injury, Roux-en-Y, pancreatectomy, anti-reflux operations, lung transplant) Gastrointestinal disorders: gastroesophageal reflux, gastric ulcer disease, gastritis, atrophic gastritis, pancreatitis Diseases of the nervous system (such as parkinsonism) Connective tissue diseases (such as scleroderma) Unknown (“idiopathic gastroparesis”) Paraneoplastic Mesenteric ischemia Viral infections: e.g., CMV Side effect of medications Metabolic and endocrine disorders: hypothyroidism, pregnancy, uremia

**Table 2 jcm-08-01127-t002:** Symptoms and causes of dumping syndrome.

Symptoms and Causes of Dumping Syndrome
Symptoms:
Early dumping syndrome (within 1 h after meal ingestion) Gastrointestinal symptoms: Abdominal pain, epigastric fullness, diarrhea, nausea, vomiting, borborygmi, and bloating Vasomotor symptoms: desire to lie down, palpitations and tachycardia, fatigue, faintness, syncope, perspiration, headache, light-headedness, hypotension, flushing, and pallor
Late dumping syndrome (1–3 h after meal ingestion) Neuroglycopenia: fatigue, weakness, confusion, hunger and syncope Autonomic reactivity: perspiration, palpitations, tremor and irritability
Causes:
Surgery: gastrojejunostomy, antrectomy, pylorectomy, pyloroplasty, esophagectomy, vagotomy, Roux-en Y bypass, Nissen fundoplication Not-surgery related: diabetes mellitus, viral illness, unkown (“idiopathic”)

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
