# Peer review of "Gastroparesis and Dumping Syndrome: Current Concepts and Management"

_jcm, 2019, doi:10.3390/jcm8081127_

Reviewer 1 Report

The authors write a and succinct review of both gastroparesis and dumping syndrome, two disorders that I feel may be more closely related than we suspect, despite opposite pathophysiologic mechanisms. This review is similar to others in the literature and sound science is presented. I have a few comments/suggestions as below.

1.    For gastroparesis prevalence numbers, I would at least mention the mayo study in Olmstead County you have cited, as this is a commonly cited study. The prevalence of definite gastroparesis here was much lower at 24.2/100,000, with definite, probable, or possible gastroparesis at 50.5/100,000 (reference #6). 

2.    It may be worth mentioning as well that gastroparesis is likely significantly under diagnosed as per reference #7. Estimated prevalence here is 1.8%. I’m a little unclear where the authors acquired their prevalence range of 1.5-3.0 %.

3.    Would mention that reliable population prevalence data for dumping syndrome is not available. At least I am not aware of any.

4.    In table 1: I would add lung transplant to list of surgical conditions that may cause gastroparesis. Also, do the authors have evidence that GERD and atrophic gastritis can cause gastroparesis. One sometimes attributes refractory GERD to gastroparesis, but I have never heard of this relationship running the opposite way. I would also remove constipation, diarrhea, and fecal incontinence from the list of gastroparesis symptoms. Motility disorders often run together, which I very much appreciate, but these symptoms are almost always due to the presence of another concurrent syndrome.

5.    In line 162, please provide a different reference, particular for the 3 month criteria.

6.    Under gastroparesis diagnostic evaluations, I would change the statement about gastric emptying scintigraphy having limited availability (line 167). In the US this is the most widely used method with breath testing barely used. Can mention regional variability though.

7.    The paragraphs on treatments for gasroparesis and dumping syndrome should be split up to make it easier to read.

8.    For line 211, change several trials to “several randomizedtrials”.

9.    In lines 225-227, Please mention that pyloric botox has been shown to notbe effective in the only 2 randomized controlled trials done on the subject (Though an argument can be made these studies are under powered) (Pubmed ID: 17944739, 18070232)

10.Aprepitant can also be considered for use in gastroparesis based on a trial with mixed effects. Though it did not meet the primary endpoint, aprepitant reduced GCSI and nausea/vomiting subscores (PMID: 29111115). Many people consider it clinically useful.

Author Response

Reviewer 1: 

General comment: The authors write a and succinct review of both gastroparesis and dumping syndrome, two disorders that I feel may be more closely related than we suspect, despite opposite pathophysiologic mechanisms. This review is similar to others in the literature and sound science is presented.

Answer: We would like to thank reviewer 1 for this very positive feedback.

Major

Comment 1: For gastroparesis prevalence numbers, I would at least mention the mayo study in Olmstead County you have cited, as this is a commonly cited study. The prevalence of definite gastroparesis here was much lower at 24.2/100,000, with definite, probable, or possible gastroparesis at 50.5/100,000 (reference #6).

Answer: We have included the prevalence numbers from the Mayo study in the epidemiology paragraph.

Comment 2: It may be worth mentioning as well that gastroparesis is likely significantly under diagnosed as per reference #7. Estimated prevalence here is 1.8%. I’m a little unclear where the authors acquired their prevalence range of 1.5-3.0 %.

Answer: We thank reviewer 1 for this important remark. We highlight now that gastroparesis is likely under diagnosed as per reference 7 with an estimated prevalence of 1.8%. We put the number now into context and give more background to the readership.

Comment 3: Would mention that reliable population prevalence data for dumping syndrome is not available. At least I am not aware of any.

Answer: We mention now that reliable population prevalence data for dumping syndrome are currently still lacking.

Comment 4: In table 1: I would add lung transplant to list of surgical conditions that may cause gastroparesis. Also, do the authors have evidence that GERD and atrophic gastritis can cause gastroparesis. One sometimes attributes refractory GERD to gastroparesis, but I have never heard of this relationship running the opposite way. I would also remove constipation, diarrhea, and fecal incontinence from the list of gastroparesis symptoms. Motility disorders often run together, which I very much appreciate, but these symptoms are almost always due to the presence of another concurrent syndrome.

Answer: We added lung transplant to table 1. As we do not have evidence that GERD or atrophic gastritis causes gastroparesis, we did not include these two disorders as possible etiologic factors. Constipation, diarrhea and fecal incontincence have been removed from the list with gastroparesis symptoms.

Comment 5: In line 162, please provide a different reference, particular for the 3 month criteria.

Answer: We provide a different reference now.(1)Since the 3 months time period is not uniformly required for diagnosis of gastroparesis, we removed this criterion from the definition.

Comment 6: Under gastroparesis diagnostic evaluations, I would change the statement about gastric emptying scintigraphy having limited availability (line 167). In the US this is the most widely used method with breath testing barely used. Can mention regional variability though.

Answer: This is a very important comment. We have omitted the statement about the limited availability of gastric emptying scintigraphy. Since it is very infrequently used in our clinical practice in Switzerland, we highlight that there are some regional differences.

Comment 7: The paragraphs on treatments for gastroparesis and dumping syndrome should be split up to make it easier to read.

Answer: The paragraphs on treatment (gastroparesis and dumping) have been split up now per reviewer 1’s suggestion.

Comment 8: For line 211, change several trials to “several randomized trials”.

Answer: This has been corrected now.

Comment 9: In lines 225-227, Please mention that pyloric botox has been shown to not be effective in the only 2 randomized controlled trials done on the subject (Though an argument can be made these studies are under powered) (Pubmed ID: 17944739, 18070232)

Answer: We mention now that two RCTs did NOT show an effect of pyloric botox. (2, 3)

Comment 10: Aprepitant can also be considered for use in gastroparesis based on a trial with mixed effects. Though it did not meet the primary endpoint, aprepitant reduced GCSI and nausea/vomiting subscores (PMID: 29111115). Many people consider it clinically useful.

Answer: We discuss now the use of aprepitant as a possible alternative treatment for gastroparesis.(4)

Reviewer 2 Report

In this review article, the authors present an overview for gastroparesis and dumping syndrome, and they touch base the current and potential therapeutic options. Overall, the review is well written and organized, and covering the scientific ground of both disorders. I have some comments for the authors:

1. I think imaging-based diagnosis methods are not covered enough. I would not state that the availability of gastric emptying scintigraphy is limited. In addition, it is important to talk about MRI based assessments as well. (Pubmed ID: 20585784)

2. The review lacks the relation between gastric electrical activity (slow waves) and these two disorders (especially gastroparesis). In Diagnostic Evaluations section, I would mention the significance of slow wave measurements as a potential diagnostic tool. (Pubmed IDs: 22643349, 26839980)

3. In Line 227-232, Please mention that gastroparesis symptoms and gastric emptying were improved with high-frequency GES. (Pubmed ID: 27284964)

4. The manuscript refers to two figures (Line 93 and Line 194) but they are missing.

Minor comments:

- Line 44: "Definitions" is misspelled in the section title.

- Line 134: type II and I -> type 2 and 1 (for consistency thru the paper)

- Both Table 1 and 2 need some editing and organization. For instance Table 1 presents 'Symptoms and causes of gastroparesis' and I think it is more accurate to include 'Diabetes' as a cause rather than listing 'Diabetic gastroparesis' as a cause.

Author Response

Reviewer 2:

General comment: In this review article, the authors present an overview for gastroparesis and dumping syndrome, and they touch base the current and potential therapeutic options. Overall, the review is well written and organized, and covering the scientific ground of both disorders.

Answer: We would like to thank reviewer 2 for this excellent feedback.

Major

Comment 1: I think imaging-based diagnosis methods are not covered enough. I would not state that the availability of gastric emptying scintigraphy is limited. In addition, it is important to talk about MRI based assessments as well. (Pubmed ID: 20585784)

Answer: This is a very important comment. We kindly refer to our answer to comment 1 of reviewer 6. We have included MRI as a potential diagnostic tool.(5)

Comment 2: The review lacks the relation between gastric electrical activity (slow waves) and these two disorders (especially gastroparesis). In Diagnostic Evaluations section, I would mention the significance of slow wave measurements as a potential diagnostic tool. (Pubmed IDs: 22643349, 26839980)

Answer: Per reviewer 2’ suggestion, we have included now gastric electrical activity (slow waves) in both the paragraphs about pathophiology as well as diagnostic evaluations.(6, 7)

Comment 3: In Line 227-232, Please mention that gastroparesis symptoms and gastric emptying were improved with high-frequency GES. (Pubmed ID: 27284964)

Answer: We mention now that gastroparesis symptoms and gastric emptying are improved with GES.(8)

Comment 4: The manuscript refers to two figures (Line 93 and Line 194) but they are missing.

Answer: We apologize for not having uploaded the two figures. This has been done now with the re-submission of the manuscript.

Minor

Comment 1: Line 44: "Definitions" is misspelled in the section title.

Answer: This mistake has been corrected.

Comment 2: Line 134: type II and I -> type 2 and 1 (for consistency thru the paper)

Answer: This has been changed according to reviewer 2’s suggestion.

Comment 3: Both Table 1 and 2 need some editing and organization. For instance Table 1 presents 'Symptoms and causes of gastroparesis' and I think it is more accurate to include 'Diabetes' as a cause rather than listing 'Diabetic gastroparesis' as a cause.

Answer: Table 1 and 2 have been edited.

Round  2

Reviewer 1 Report

Authors have addressed all suggestions.

Reviewer 2 Report

Have no further comments.